# What We Should Consider in Point of Care Blood Glucose Test; Current Quality Management Status of a Single Institution

**DOI:** 10.3390/medicina57030238

**Published:** 2021-03-04

**Authors:** Sooin Choi, Soo Jeong Choi, Byung Ryul Jeon, Yong-Wha Lee, Jongwon Oh, You Kyoung Lee

**Affiliations:** 1Department of Laboratory Medicine and Genetics, Soonchunhyang University College of Medicine, Soonchunhyang University Bucheon Hospital, 170 Jomaru-ro, Bucheon 14584, Korea; sooin2@schmc.ac.kr (S.C.); unifi@schmc.ac.kr (B.R.J.); lywmd@schmc.ac.kr (Y.-W.L.); 2Department of Internal Medicine, Soonchunhyang University College of Medicine, Soonchunhyang University Bucheon Hospital, 170 Jomaru-ro, Bucheon 14584, Korea; crystal@schmc.ac.kr; 3Department of Laboratory Medicine, Soonchunhyang University College of Medicine, Soonchunhyang University Cheonan Hospital, 31 Soonchunhyang6gil, Cheonan 31151, Korea; tltcjg@naver.com

**Keywords:** POCT, glucometer, blood glucose test, education, quality control, process, post-analytical error

## Abstract

*Background and Objectives:* Point of care test (POCT) is generally performed by non-laboratory staff who often lack an understanding on the quality control and quality assurance programs. The purpose of this study was to understand the current status of quality management of point of care (POC) blood glucose testing in a single institution where non-laboratory staff perform the tests. *Materials and Methods:* From July to August 2020, management status of glucometer, test strips, quality control (QC) materials, quality assurance program, and operators’ response to processing of displayed results was monitored in all Soonchunhyang University Bucheon hospital departments that performed POC blood glucose test. Results of the POC blood glucose test conducted from January 2019 to May 2020 were analyzed retrospectively. *Results:* A total 124 glucometers were monitored in 47 departments. Insufficient management of approximately 50% of blood sugar, test strips, and QC materials was observed. Although daily QC was conducted by 95.7% of the departments, the QC records were inaccurate. The method of recording test results varied with departments and operators. Various judgments and troubleshooting were performed on the unexpected or out of measurable range results, including some inappropriate processes. In POC blood glucose test results review, 4568 atypical results were identified from a total of 572,207 results. *Conclusions:* Sufficient training of the non-laboratory staff and ongoing assessment of competency through recertification is needed to maintain acceptable levels of POCT quality. In this study, various problems were identified in glucometer and reagent management, QC and post-analytic phase. We believe that these results provide meaningful basal information for planning effective operators’ training and competency evaluation, and the development of an efficient POCT quality management system.

## 1. Introduction

Point of care testing (POCT) is a diagnostic test conducted at a place where care is conducted, not at the central laboratory. POCT can quickly report results, which can be quickly applied to diagnosis and treatment of patients, contribute to improved patient satisfaction by reducing blood sample volume, and meet the needs of clinicians who need to treat more patients [1]. In hospitals, control of blood glucose in narrow therapeutic ranges plays an important role in preventing complications and lowering the mortality rates attributable to glycemic fluctuations. While central laboratory testing of plasma glucose remains the standard reference, POC blood glucose test is used widely at the patient bedside because of its reliable quantitative result, portable size, low cost and simple operation [2,3]; it enables immediate determination of glucose levels, and fast treatment decisions in response to glycemic fluctuations in hospitalized patients. However, like with any laboratory test, errors can occur at any point in the testing cycle of POC blood glucose test; more so because POCT is performed by clinical staff rather than laboratory trained individuals which can lead to errors resulting from a lack of understanding of quality control and quality assurance program. In fact, operator competency has been reported as a major factor contributing to the poor analytical performance of POC blood glucose test [4].

Since 2001, our institution has distributed blood glucose test manual and conducted operator training every year. In 2017, several cases of adverse event attributed to POC blood glucose test were reported in our institution; such as hypoglycemia caused by insulin administration to false increased blood sugar levels. In response, the POC blood glucose test manual including the test procedure and quality control (QC) method was revised in 2018 (as shown in Figure 1) and the medical device safety information monitoring center in the institution plans a quality improvement program. Before the program was implemented, the need to identify problems emerged. Although there were several reports related to quality management of POC blood glucose tests, the following points that can be seen as differences in this study. (1) Understanding the current status of the operator’s manage sequence to patient outcomes, (2) check QC log accuracy, (3) review of test results recorded using unappointed symbols (atypical record) and (4) check the cleaning of the equipment. This study was conducted to identify the current status of quality management of POC blood glucose test performed by non-laboratory staff, and to identify which parts are vulnerable and share them as basic data for POC blood glucose test quality improvement.

## 2. Materials and Methods

Monitoring and retrospective data analysis were conducted at Soonchunhyang university Bucheon hospital in Bucheon, Gyeonggi Province, Republic of Korea. This study was approved by the Soonchunhyang University Bucheon Hospital (SCHBC 2019-09-006-002, approved in October 2019) Institutional Review Board and written informed consent was waived.

### 2.1. Monitoring

Monitoring was conducted in all departments that conducted POC blood glucose test from July to August 2020. Two observers monitored each department independently and the results were compared, and in case of discrepancies, a third observer monitored the department again to determine the final result. The items that were monitored were:Glucometer: Test Strip Inlet Cleaning, Measurement Optical Window Cleaning, and Storage.

To evaluate the effectiveness of the cleaning process, the glucometer, test strip inlet, and measurement optical window of individual devices, were cleaned with one sheet of alcohol swab, two observers simultaneously observed the results through naked eyes and compared the colors of the unused and used alcohol swab (Figure 2). Glucometer was defined as either “clean” or “not-clean” depending on the results given by the two observers. Only the glucometers which were determined by the two observers as clean were defined as “clean”. The glucometers that one or more observers determined as discolored were defined as “not clean”.

2.Test strip kit: serial and lot number of kits, opening date, expiration date, and sealing status were checked. Only the test strip kit, which did not move at all when the lid was pressed down, was determined as “sealed”.3.QC material: Opening date and expiration date.4.Quality assurance program: recording of information about strip and QC material including recording format, review QC and troubleshooting results.5.Others: type of samples, sampling method, blood applying method and result recording.

### 2.2. Manage Sequence to Specific Results

Manage sequence taken by operators in three different situations; unexpected results, above and below the measurable range (displayed as Hi and Lo, respectively). After the preliminary monitoring, we created 10 choices for the unexpected results, 7 choices for results above or below the measurable range. For the wards, each department was divided into two to five teams and blood glucose tests were conducted, so monitoring was performed on each team. Multiple choices for each situation were checked.

### 2.3. Glucose Level Data Analysis

Accu-chek Active blood glucometer and test strips (Roche Diagnostics, Mannheim, Germany) are used in the entire institution. The meter used glucose dehydrogenase method and measurement is done from the bottom of the strip on the color comparison field by reflectance photometry. Measurable range was 10 to 600 mg/dL. Independent menu (which is then named blood sugar test (BST menu) for recording manual POC blood glucose test results was provided in the electronic health record system. The data analysis included only the data entered on this menu. In the preliminary analysis of the BST menu data, results out of measurable range, results containing decimal points, or symbols were analyzed.

When the results had decimal points, symbol and letter, we classified as “atypical BST results”. The percentage of these results for the entire BST menu data was calculated.The retrospective data of blood glucose test of atypical BST results was compared with the results of the central laboratory or POC glucometer received approximately three minutes before or after the time the atypical BST result was recorded.Each department team showed examples of atypical BST results and asked if they had ever recorded the results and what they meant.

In the central laboratory, glucose was measured by the hexokinase glucose 6-phosphate dehydrogenase method using L-type Glu 2 (Wako Pure Chemical Industries, Osaka, Japan) in a Hitachi 7600-110 analyzer (Hitachi Ltd., Tokyo, Japan). In POC gas analyzer, GEM Premier 5000 (Instrumentation Laboratory, Bedford, MA, USA) provides quantitative measurements of whole blood glucose. Glucose determination is accomplished by enzymatic reaction of glucose or lactate with oxygen in the presence of glucose oxidase or lactate oxidase and the electrochemical oxidation of the resulting hydrogen peroxide at the platinum electrode.

### 2.4. Statistical Methods

Categorical data are summarized as count and percent. In continuous data, normality test using the Shapiro-Wilk test was conducted and data summarized as mean in normal distribution or median in non-normal distribution. Statistical analyses were performed using Analyse-it v5.10 (Analyse-it Software, Leeds, UK) and Microsoft Excel 2019 (Microsoft, Redmond, WA, USA).

## 3. Results

General characteristics of the institution and the departments involved in this study are shown in Table 1. A total of 127 glucometers were distributed to 50 departments. The average daily test strip usage of our institution was 1403 from January 2019 to May 2020. Surgery outpatient clinic, dental outpatient clinic, and angiography departments were excluded from monitoring, because there was no POC blood glucose test prescription in these departments during the monitoring period. Finally, 124 glucometers in 47 departments (26 wards, 11 outpatient clinics and 10 special departments) were monitored.

### 3.1. Monitoring

Blood glucose test monitoring results were shown in Table 2.

#### 3.1.1. Glucometer

The temperature (mean 24.77 °C, SD 1.41) and humidity (median 33.0%, interquartile range, IQR 30.0–36.8) of all storage locations were distributed within the appropriate range indicated in the device manual.

#### 3.1.2. Test Strip Kit

The number of test strip kits opened and in use was 107 and median was 2 (IQR 1.0–3.0). Total 11 lot types were in use, and three of lot types were recommended for disposal because they were expired. Each department was using single lot types of test strips (IQR 1.0–2.0). The total number of strip kits in stocks was 305, with eight lot types. The median and maximum number of the total number of lot types in each department was 2.0 and 5.

#### 3.1.3. QC Material

A total of 48 QC materials were opened and in use, only 45.8% (22/48) of them had records of opening date.

#### 3.1.4. Quality Assurance Program

A total of 45 (95.7%) departments carried out daily QC and their recording status was as Figure 3.

#### 3.1.5. Others

The most commonly used sample was the capillary blood, used by 97.9% (46/47) of departments, followed by venous blood and arterial blood. Five departments used cerebrospinal fluid (CSF) and one department used urine as the sample. A majority of the departments used squeezing methods for the collection of capillary blood (91.5%, 43/46). A majority of the departments used finger contact in the glucometer to apply capillary blood with the test strip. When recording results on electronic health record (EHR), 59.6% (28/47) of departments used BST menu and 36.2% used nursing record (Figure 4). Two departments used either menus depending on each user. The departments that immediately entered the results displayed on the glucometer into the EHR using a computer at the testing site were 38.3% (18/47). 48.9% (23/47) of departments printed out the worklist of each ward registered in the EHR, recorded the results by manual, and entered it in EHR after coming to the station. Others, 12.8% (6/47) used note paper, and not the worklist.

### 3.2. Manage Sequence to Specific Results

A total of 95 departments or teams participated, but some respondents said they did not know how to deal with each situation (Figure 5).

Unexpected values response: More than 80% of respondents said they would check the patient’s condition (93.5%, 86/92) and report it to the clinician (88.0%, 81/92) if unexpected results were displayed.Measurable range values response: The most common response was to notify the clinician (75.4%, 52/69) followed by retesting (65.2%, 45/69).Measurable range values response: The most common response was to notify the clinician (70.8%, 51/72) followed retest (68.1%, 49/72).

### 3.3. Glucose Level Data Analysis

The total number of test strip ordered during the analysis period was 768,450 (Table 3). The BST menu dataset contained 572,027 results for 13,786 patients from 23 departments. A total of 4568 atypical BST results were identified. Twenty-two of them were out of measurable range (20 results above 600 mg/dL and 2 results below 10 mg/dL), and the remaining 4546 results were recorded by symbol that could not be discussed alone, or by combination of numbers and symbols. The main types of symbols used were “.”, “,” “-”, “--,” and “---”.Twenty-two atypical BST results out of measurable range were compared with the results of the central laboratory or POC glucometer. For those below the measurable range, no other tests were performed simultaneously. A total of 10 results were above the measurable range results, two were identical to serum glucose results and seven were identical to arterial blood gas analysis (aBGA) results. One of the serum glucose tests was performed simultaneously, but the results were different with atypical BST results recorded on BST menu.Although atypical BST results were observed in 23 departments, only 10 teams among 64 teams belongs to previous 23 departments were aware of the meaning of the results recorded as symbols. In all the departments that did not report atypical results no one knew the meaning of the symbols. As for the meaning of the symbol, the most frequent answer was “not tested”.

## 4. Discussion

In this study, operators were found to be non-compliant with maintenance and QC measures, and POC blood glucose test was non-compliant with the laboratory standards institute (CLSI) guideline or manufacturer’s manual.

Basic maintenance of which includes cleaning and disinfection was poorly; 46.0% of the test strip inlets and 55.6% of measurement optical windows were not clean (Table 2). Accu-chek Active detects color change of test strip with reflectance photometry when measuring the glucose level. Measurement optical window contamination can interfere the detection of color changes and affect the results. Moreover, previous reports about the infection control issues related to glucose meter testing reported serious cases such as acute hepatitis B virus transmission which was attributed to unsafe practices among them, sharing of blood glucose meters between patients without cleaning or disinfecting them in between tests [5,6]. Inappropriate cleaning was determined to create synergy with capillary blood sampling method to increase infection risk; 95.7% of departments used the method touched the patient’s fingers with the glucometer. According to the manufacturer’s manual, if a single device is to be used by more than one patient, blood should be applied while the test strip is outside the device; remove the test strip from the device and insert after applying the blood sample. Cleaning and disinfection of glucometer should be performed to prevent transmission between patients [7] and has to be a routine practice after each use, regardless of whether it has been shared between patients [8].

Temperature, and humidity are common factors that affect the glucometer measurements [9]; the enzymes on the strip can be inactivated at extreme temperature while exposure to humidity can prematurely rehydrate the enzyme and limit its reactivity during for patient testing [10]. In this study, the temperature and humidity of all departments met the manufacturer’s suggestion. However, 55.1% of the monitored test strip kits in use were not sealed and exposure to moisture could not be ruled out. The problem of test strip lot management was also identified. There is often a lot-to-lot test strips variation on glucose results, which affects the accuracy of the results [11]. Therefore, variation verification of each lot of strips before use is necessary [12], and a large number of sequestered strips are usually ordered in order to avoid frequent change of strip lots. In our institution, however, eleven and eight lot types were identified in the test strip in use and in stock, respectively and parallel tests were not performed when test strips lot was changed.

In the present study, several QC problems were identified. First, The QC material is valid for two months from the opening date as indicated in the user manual, in our institution, 45.8% of the departments did not specify the opening date, making it difficult to determine whether the reagent was suitable for use. Second, although 95.7% of departments performed QC every day (or before test), all departments were unaware of the additional QC in addition to daily QC. Additional QC is required in accordance with clinical and Clinical and Laboratory Standards Institute (CLSI) guidelines and manufacturer’s instructions in the following cases; test strip lot change, new test strip kit is opened, after batteries are replaced, whenever questionable results are obtained, and after cleaning/disinfecting the test strip inlet and measurement optical window [7,13,14]. In previous study conducted in the United Kingdom observed that 2% of the centers never performed QC, only 29% performed QC when starting a new batch of test strips, and only 15% performed QC as per recommended guidelines [15]. Third, QC documentation was inaccurate and the review process was not properly conducted. In QC log, 73.3% and 40.0% of departments did not include the test strip and QC material lot number, respectively. And even when the test strip and QC material lot numbers were documented, 67.0% and 26.0%, respectively, were mismatched with the actual lot. It was determined that training on QC documentation methods would be required. All QC results should be recorded in the QC log and any corrective action should be carefully recorded in the log. In this study, 74.5% of the departments performed retest as a QC troubleshooting, but only 8.5% of the departments recorded initial result and 29.8% of the departments recorded retest result.

Just like any laboratory test, errors can occur at any point in the testing cycle of POC blood glucose test. Previously, in pre-analytical phase, a higher rate of errors such as patient identification was found in POCT compared to central laboratory testing [16]. In analytical phase, delay in testing due to an operator not being certified to perform testing was reported as the most common error related to POCT glucose test [17]. In the post-analytical phase, POCT may enjoy some advantages, because the clinician is immediately available to view it [18]. However, the possibility of errors remains. Data from small devices and disposable devices used in POCs are more likely to be recorded manually, and our institution was still entering POC blood glucose results manually. In the present study, several problems related to the transcription of the results were found. First, multiple EHR menus for recording results (Figure 4) were used. Recording results on multiple menus can make it difficult for clinicians to identify consecutive blood sugar levels of each patient. When POCT results are not available in the patient chart used by clinician, repeat testing is often performed which leads to increased costs and patient discomfort [19]. Second, some of the results were determined as clear record error such as results out of measurable range, and results indicated by symbols (Table 3), although this study did not perform a comparative assessment with the true value. Manual transcription errors occurrence were previously reported at rate of approximately 3% [20,21]. To avoid inconsistencies with manual recording, there should be specific standards on where and how POCT results will be charted. May et al. suggested that user interface improvements could prevent a portion of errors; requiring only numeric characters, results in a physiologically possible range, or double entry of all manually entered results [22]. Third, the results obtained through various samples or test methods were recorded without any particular distinction. Blood glucose level is measured differently depending on the sample. Results from sample other than capillary blood should be identified, and there should be clear distinction between the test results performed using POC glucometer and other methods [14]. Test results using unverified types of samples, such as CSF and urine, cannot guarantee test quality, so it is recommended that POC blood glucose test should not be performed using these samples.

The operator’s response to unexpected results or results out of measurable range was varied and included some of the actions which is unnecessary by CLSI guidelines (Figure 5, gray boxes). When unexpected result is displayed, the operator is advised to repeat the test before taking action [13]. In this study, various retest methods were used; using same/different glucometer and same/different test strip. 42.4% (39/92) and 51.1% (47/92) of responders retested the used test strip by inserting it back into the same or other glucometer, respectively. But test strips should not be reused. Reviewing the proper test procedure and checking for possible meter damage is recommended [13]. For out of measurable range results, samples should not be repeated on a POC glucometer [14]. Because of criticism regarding the accuracy of POC blood glucose test at very high and low results, most glucometers used today in hospitals do not display results if it is either less than or greater than the measurement range. For instance, Accu-chek Active display these results as word “Lo” or “Hi”. Typical protocols include that operator should obtain a specimen and send it for definitive testing in central laboratory [8] and notify the clinician.

Operator competency is important to POCT quality because it is one of the potential sources of error [23]. POCT is generally performed by members of the clinical team who oftenly lack an understanding on the quality control and quality assurance program. Although the model of a satellite POCT laboratory staffed by Medical Laboratory Technologists suggested as an alternative to POCT performed by clinical staff [24], an additional cost of satellite laboratory would be inhibitory to its implementation. In this situation, sufficient training to the non-laboratory staff and ongoing assessment of competency through recertification [25] is the first step toward reducing the incidences of human error [26]; training improved operators’ competence and precision of POC blood glucose test [27,28]. Operators should be trained not only on how to use the device, but also on how and where to chart results, how and why quality control is performed and troubleshooting [25,29]. With many operators to train and limited resources to support POCT programs, online training becomes an attractive option [25]. Evaluation process for operators’ capabilities is essential; Tang et al. presented a checking list of glucose meter operation competence assessments as a useful tool [30].

To improve the quality management program of POC blood glucose test, our institution make the following plans. (1) Add POC blood glucose test related detailed manual: manage sequence to displayed results, various cases required QC, and parallel test. (2) Training on detailed manual. (3) Regular operator test and on-site monitoring after training. (4) Daily management checklist for devices, test strips and QC materials. (5) Introduction of glucometer that can be connected to EHR and automatically upload results in order to solve the problem of recording results in various ways by department and to prevent recording error [31,32].

The main limitation of this study was small sample size because it was performed at a single institution, and error identification was not performed by comparing them with the true values. In the future, for improved accuracy, study using larger and heterogeneous population would be required.

## 5. Conclusions

This study showed the status of POC blood glucose test quality management and manage sequences to test results of non-laboratory staff. Various problems have been identified in glucometer and reagent management, QC and result management. We believe that these results provide meaningful basal information for planning effective operators training and competency evaluation, and development of an efficient POCT quality management program.

## Figures and Tables

**Figure 1 medicina-57-00238-f001:**
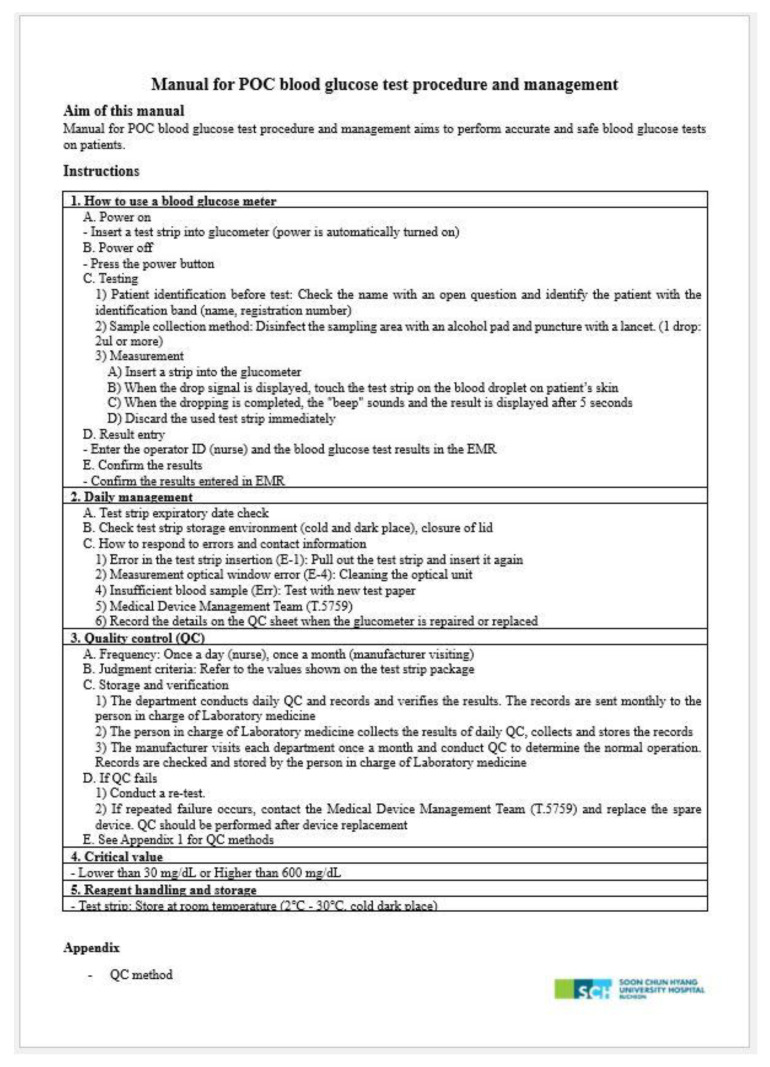
The point of care (POC) blood glucose test manual revised in 2018. Soonchunhyang university Bucheon hospital has distributed this manual and conducted operator training every year.

**Figure 2 medicina-57-00238-f002:**
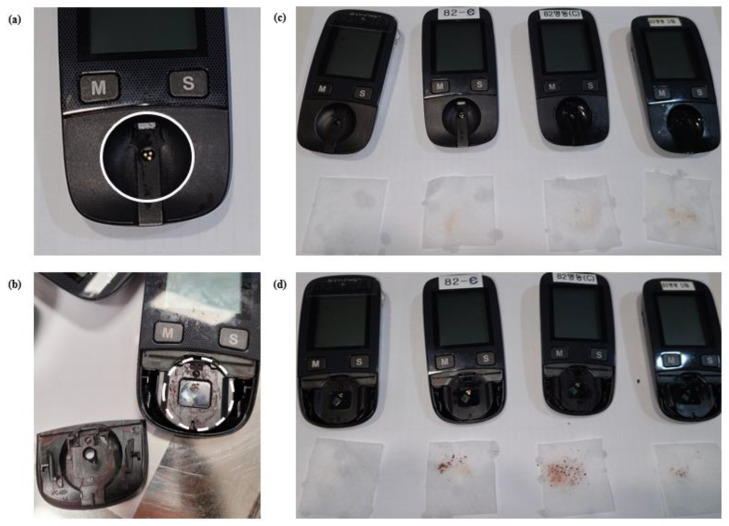
The test strip inlet ((**a**) white circle) and measurement optical window ((**b**) white dotted circle) exposed when cover is removed. Evaluating the effectiveness of glucometer cleaning method; test strip inlet and the measurement optical window of individual equipment were cleaned with one sheet of alcohol swab for, two observers simultaneously observed the results using naked eyes and compared the colors of unused and used alcohol swab. (**c**) Test strip inlet cleaning; all four glucometers were defined as “not clean”. (**d**) Measurement optical window cleaning; all four glucometers were defined as” not clean”.

**Figure 3 medicina-57-00238-f003:**
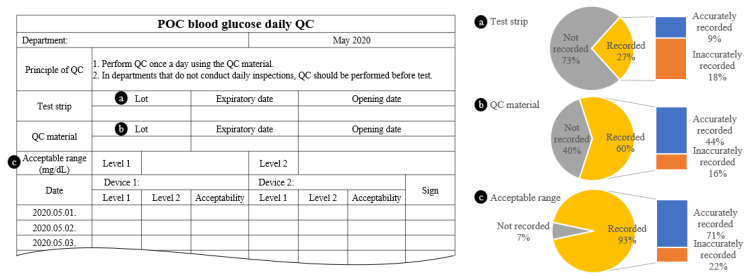
Quality control (QC) log format and the recording status of 47 departments; (**a**) Test strip, (**b**) QC material, and (**c**) acceptable range.

**Figure 4 medicina-57-00238-f004:**
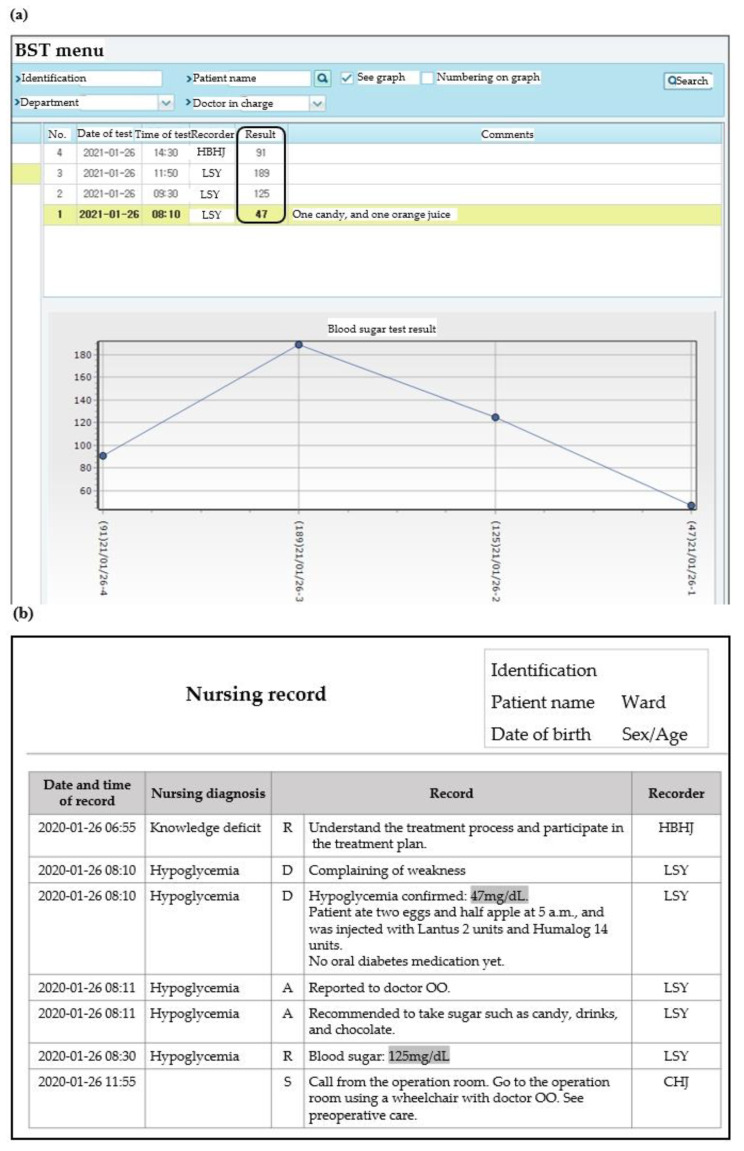
Two electronic health record menus used for point of care (POC) blood glucose test results recording: (**a**) independent menu for recording manual POC blood glucose test results (named blood sugar test (BST) menu) and (**b**) nursing record. The POC blood glucose test results (shaded in gray) recorded in the nursing record menu were mixed with various other records, making it difficult to identify patients’ serial blood glucose levels.

**Figure 5 medicina-57-00238-f005:**
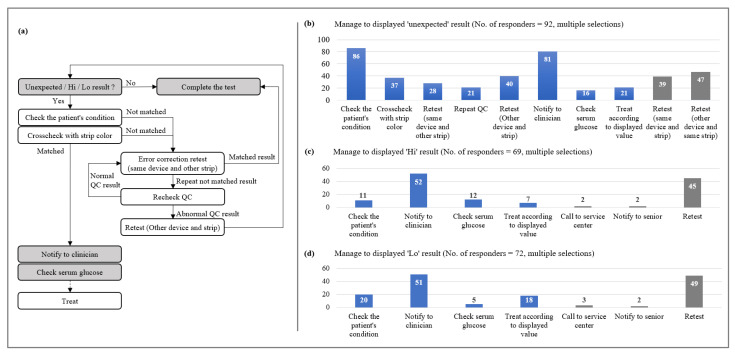
Manage sequence for the results displayed on the glucometer; (**a**) flowchart of how to respond to unexpected, above the measurable range (displayed as Hi) or below the measurable range (displayed as Lo). Multiple selection results of (**b**) 92 responders for unexpected results, (**c**) 69 responders for Hi results, and (**d**) 72 responders for Lo results. Among the selections, the manage sequences not included in the flowchart (which the authors judged to be inappropriate or unnecessary) were presented in gray boxes (**b**–**d**).

**Table 1 medicina-57-00238-t001:** General characteristics of the institution and distribution of departments involved in this study.

**General Characteristics of the Institution**		
Number of beds		928
Average daily outpatients		3093
Average daily inpatients		801
Total number of glucometers		127
Number of glucometers monitored in this study		124
Average daily test strip usage in 2019		1403
Total number of departments performing blood glucose tests		50
Number of departments monitored in this study		47
**Distribution of Departments Involved in This Study**		
Ward (*N* = 26)	Outpatient clinic (*N* = 11)	Special department (*N* = 10)
Delivery room Twenty general wards Four intensive care units Neonates’ room	Allergy and respiratory medicine Cardiology Comprehensive check-up center Endocrinology Family medicine Gastroenterology General check-up center Nephrology Oncology and hematology Pain medicine Pediatrics	Bronchoscope Cardiovascular center Central injection CT preparation Day operation center Dialysis Emergency room Nuclear medicine injection Nursing department Operating room

**Table 2 medicina-57-00238-t002:** Results of blood glucose test monitoring conducted in 47 departments.

**Glucometer**							
Total no. of device		124					
Cleaning	Test strip inlet	Clean		Not clean			
		67	54.0%	57	46.0%		
	Measurement optical window	Clean		Not clean			
		55	44.4%	69	55.6%		
**Test Strip Kit**							
Opening date notation		Yes		No			
		3	2.8%	104	97.2%		
Sealing		Sealed		Not sealed			
		48	44.9%	59	55.1%		
**QC Material**							
Opening date notation		Yes		No			
		26	54.2%	22	45.8%		
**Other**							
Types of samples used (duplicated check)		Capillary blood *		Venous blood		Arterial blood	
		46	97.9%	31	66.0%	8	17.0%
		CSF		Urine			
		5	10.6%	1	2.1%		
Capillary blood sampling method		No squeezing		Squeezing			
		3	6.4%	43	91.5%		
Test strip and lancing device deposition		Bring disposal container		Using tray			
		44	93.6%	3	6.4%		

* In the Dialysis department, only arteriovenous fistula sample were used. SD: standard deviation, IQR: interquartile range, No.: number, QC: quality control, CSF: cerebrospinal fluid.

**Table 3 medicina-57-00238-t003:** Results of blood glucose test are recorded differently for each department, and within the same departments. Some departments entered results outside the measuring range, results containing decimal points, or symbols, but only a few operators were aware of the symbols meaning.

Department	Approximate Test Strip Usage	No. of Results			No. of Respondents		
		Total no. on BST Menu	Atypical	(%)	Total	Who Know the Meaning of the Symbol	(%)
**Ward**							
Delivery room	1700	1126	0	0.0%	1	0	0.0%
General ward 1	36,750	31,168	170	0.5%	2	0	0.0%
General ward 2	24,600	19,117	340	1.8%	4	0	0.0%
General ward 3	44,600	35,553	399	1.1%	5	3	60.0%
General ward 4	31,050	22,938	24	0.1%	5	1	20.0%
General ward 5	21,150	16,531	43	0.3%	2	0	0.0%
General ward 6	49,450	41,359	2139	5.2%	2	2	100.0%
General ward 7	35,000	26,728	897	3.4%	3	1	33.3%
General ward 8	25,550	20,578	80	0.4%	2	0	0.0%
General ward 9	36,450	27,607	19	0.1%	2	0	0.0%
General ward 10	22,100	17,255	18	0.1%	1	0	0.0%
General ward 11	20,100	16,849	143	0.8%	2	1	50.0%
General ward 12	10,000	3811	14	0.4%	2	0	0.0%
General ward 13	28,350	20,667	7	0.0%	2	0	0.0%
General ward 14	29,500	21,165	28	0.1%	2	1	50.0%
General ward 15	26,200	21,115	7	0.0%	4	0	0.0%
General ward 16	45,600	34,433	11	0.0%	3	0	0.0%
General ward 17	28,750	19,990	23	0.1%	6	0	0.0%
General ward 18	25,900	19,042	0	0.0%	4	0	0.0%
General ward 19	19,800	15,082	71	0.5%	2	0	0.0%
General ward 20	29,650	23,851	12	0.1%	2	0	0.0%
ICU 1	43,450	41,547	27	0.1%	7	1	14.3%
ICU 2	29,000	28,823	20	0.1%	1	0	0.0%
ICU 3	37,750	37,086	47	0.1%	2	0	0.0%
NICU	6300	1831	0	0.0%	4	0	0.0%
Neonates’ room	5150	3522	29	0.8%	1	0	0.0%
**Outpatient clinic**							
Allergy and respiratory medicine	150	0	0	NA	1	0	0.0%
Cardiology	1300	0	0	NA	1	0	0.0%
Comprehensive check-up center	100	0	0	NA	1	0	0.0%
Endocrinology	4950	0	0	NA	1	0	0.0%
Family medicine	150	0	0	NA	1	0	0.0%
Gastroenterology	150	0	0	NA	1	0	0.0%
General check-up center	800	0	0	NA	1	0	0.0%
Nephrology	250	0	0	NA	1	0	0.0%
Oncology and hematology	50	2	0	0.0%	1	0	0.0%
Pain medicine	200	0	0	NA	1	0	0.0%
Pediatrics	50	0	0	NA	1	0	0.0%
**Special department**							
Bronchoscope	200	0	0	NA	1	0	0.0%
Cardiovascular center	50	0	0	NA	1	0	0.0%
Central injection	2850	1787	0	0.0%	1	0	0.0%
CT preparation	250	0	0	NA	1	0	0.0%
Day operation center	1550	1202	0	0.0%	1	0	0.0%
Dialysis	25,500	0	0	NA	1	0	0.0%
Emergency room	28,300	262	0	0.0%	1	0	0.0%
Nuclear medicine injection	2100	0	0	NA	1	0	0.0%
Nursing department	250	0	0	NA	1	0	0.0%
Operating room	2350	0	0	NA	1	0	0.0%
**Summary**	768,450	572,027 (13,786 patients)	4568	0.8%	94	10	10.6%

No.: number, ICU: intensive care unit, NICU: neonate intensive care unit, NA: not available, CT: computer tomography.

## Data Availability

The data for this study are available on request from the corresponding author, through institutional review board, and reviewers. The data are not publicly available due to restrictions of obtaining approval from the IRB for the data disclosure. If anyone requires our data of this study, please do not hesitate to contact the corresponding author.

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
