# Peer review of "What We Should Consider in Point of Care Blood Glucose Test; Current Quality Management Status of a Single Institution"

_medicina, 2021, doi:10.3390/medicina57030238_

Round 1

Reviewer 1 Report

This manuscript discusses the current status of quality management of POC blood glucose testing in an institution. The management status of glucose meter, test strips, QC and quality assurance program were discussed. The current issues were mentioned and potential future works were suggested. Overall, the study provides information for the development of a more efficient POC quality management system. The work is interesting. The following comments should be addressed before the paper can be considered:

  • In discussion, the authors mentioned that “Measurement optical window contamination can affect the results.” Can authors further elaborate how this happens?
  • The authors should justify why this experiment only performed for one institution.
  • In introduction, the authors should discuss the importance of POC testing in general. The authors should cite and discuss some POC related papers, followed by the discussion of the existing glucose meters according to the literature. They include but not limited to the followings:

Biosensors and Bioelectronics107, 266-271. (2018)

Sensors and Actuators B: Chemical243, 484-488. (2017)

Lab on a Chip17(3), 382-394. (2017)

  • In discussion, the authors should discuss if any improvement of current glucose meters can be done to address current challenges. For instance, automation can be implemented to the device to reduce human errors. Some relevant publications should be cited.

Author Response

  1. In discussion, the authors mentioned that “Measurement optical window contamination can affect the results.” Can authors further elaborate how this happens?
  • To help readers understand, we modified the phrase as follows: Accu-chek Active detects color change of test strip with reflectance photometry when measuring the glucose level.
    T
    he results from the contamination were not But, FDA documents and manufacturer's instructions indicate that the measurement results could be affected by measurement window contamination.
  •  
  1. The authors should justify why this experiment only performed for one institution.
  • Thank you for great comment. This study is a kind of audit report. We agreed it would be better if the same study was conducted in multi-centers. It is difficult to know if it applies to other institutions where quality management procedures have been put in place. We aimed to improve the real world of POC glucometer in health institute, not general user. For the improvement, our hospital plans to replace glucometer which enter the test results automatically through the network. This new system will solve the problems with the existing manual recording. In addition, we took user training periodically.

  1. In introduction, the authors should discuss the importance of POC testing in general. The authors should cite and discuss some POC related papers, followed by the discussion of the existing glucose meters according to the literature. They include but not limited to the followings:

Biosensors and Bioelectronics107, 266-271. (2018)
Sensors and Actuators B: Chemical243, 484-488. (2017)
Lab on a Chip17(3), 382-394. (2017)

  • Thank you for kind comment and references. We wrote a description of the importance of the POC and added references.

  1. In discussion, the authors should discuss if any improvement of current glucose meters can be done to address current challenges. For instance, automation can be implemented to the device to reduce human errors. Some relevant publications should be cited.
  • To prevent errors related to the recording of results, our institute will introduce a POC blood glucose test system in which results are automatically uploaded to the EHR. This content and related literature were added.

Reviewer 2 Report

This paper reflects the experiences of an audit of glucose monitoring by non-laboratory trained staff in a single hospital over a period of months during 2019/2020. The results of this audit showed numerous problems in the care of devices, QC procedures, reagent management, recording and understanding of results.  The paper concludes that an efficient POCT quality management system is required. The major problem is that the methods do not describe what training (if any) staff had been given to use glucometers and what QC procedures were in place before the audit was carried out. Therefore the conclusion that a quality management system is required is not surprising if no training was given in the first place. As this is the experience of one institution, it is difficult to know if it applies to other institutions where quality management procedures have been put in place, and as many of these issues have previously been reported, the paper does not add a lot to the literature. No practical guidelines are given as to what needs to be included in the quality management programme and how it would be put into place.  

The paper contains a lot of data, much of which is presented in a confusing manner and could do with some simplification. For example, Table 2 is complex and difficult to read, and the authors need to consider breaking it down or changing the format. Similarly Fig 3 is very confusing.

There are some sentences where grammar needs correction or need to be rewritten. E.g. 

P2 Line 49; “In 2017, several cases adverse events….’ Needs to be rewritten

P4 Line 120; Sentence “Finally, 124 glucometers…” needs to be completed.

Author Response

  1. The major problem is that the methods do not describe what training (if any) staff had been given to use glucometers and what QC procedures were in place before the audit was carried out. Therefore the conclusion that a quality management system is required is not surprising if no training was given in the first place.
  • Thank you for vaulable comments. We added information about our institution's manual and user training in the introduction.
  1. As this is the experience of one institution, it is difficult to know if it applies to other institutions where quality management procedures have been put in place, and as many of these issues have previously been reported, the paper does not add a lot to the literature.
  • We understand your comments on the limitations of single-institution study and think that better data could be provided if the same study was conducted with multi-institution studies. However, our hospital plans to replace glucometer which enter the test results automatically through the network, to solve the problems with the existing manual recording. Finding an institution with the similar conditions as our institution was practically difficult, because we plan to evaluate the improvement of the BST quality management by replacing devices and user training in the future.
  • Although there have been reports related to quality management of POC blood glucose tests, the following points that can be seen as differences in this study.

1) Understanding the current status of the operators' manage sequence to patient outcomes

2) Check QC log accuracy

3) Review of test results recorded using unappointed symbols (atypical record)

4) Check the cleaning of the equipment

There is a possibility that essential processes are missing from the work of non-laboratory employees for those working in the central inspection room. In particular, 1) believes that it would be useful for readers to share the monitoring experience of our institution, as site judgments and additional examinations of specific outcomes are important for patients' treatment.

  1. No practical guidelines are given as to what needs to be included in the quality management programme and how it would be put into place.  
  • We believe that the guidelines of the POC blood glucose test quality management program, as in your comments, will help the reader. However, these guidelines have been published as CLSI guidelines and is included in this manuscript's references.
  1. The paper contains a lot of data, much of which is presented in a confusing manner and could do with some simplification. For example, Table 2 is complex and difficult to read, and the authors need to consider breaking it down or changing the format. Similarly Fig 3 is very confusing.
  • Thank you for your careful comment. To prevent confusion among readers, part of table 2 is separated into figure3.
  • To reduce confusion about Figure 5 (formerly Figure 3, you mentioned), we simplified the data.
  1. There are some sentences where grammar needs correction or need to be rewritten.

P2 Line 49; “In 2017, several cases adverse events….’ Needs to be rewritten

P4 Line 120; Sentence “Finally, 124 glucometers…” needs to be completed.

  • Thank you for your careful comment. I revised the sentences.

Round 2

Reviewer 2 Report

The authors have responded to my comments and made adjustments to the manuscript